# Temporal Extrapolation and Knowledge Transfer for Lifelong Temporal Knowledge Graph Reasoning

**Zhongwu Chen**[1], **Chengjin Xu**[2*], **Fenglong Su**[3*], **Zhen Huang**[1*], **Yong Dou**[1]

[1]National Key Laboratory of Parallel and Distributed Computing,
College of Computer, National University of Defense Technology;
[2]International Digital Economy Academy (IDEA);
[3]National University of Defense Technology.
{chenzhongwu20,sufenglong18,huangzhen,douyong}@nudt.edu.cn,xuchengjin@idea.edu.cn

## Abstract

Real-world Temporal Knowledge Graphs keep growing with time and new entities and facts emerge continually, necessitating a model that can extrapolate to future timestamps and transfer knowledge for new components. Therefore, our work first dives into this more realistic issue, lifelong TKG reasoning, where existing methods can only address part of the challenges. Specifically, we formulate lifelong TKG reasoning as a temporal-path-based reinforcement learning (RL) framework. Then, we add temporal displacement into the action space of RL to extrapolate for the future and further propose a temporal-rule-based reward shaping to guide the training. To transfer and update knowledge, we design a new edge-aware message passing module, where the embeddings of new entities and edges are inductive. We conduct extensive experiments on three newly constructed benchmarks for lifelong TKG reasoning. Experimental results show the outperforming effectiveness of our model against all well-adapted baselines.

## 1 Introduction

Knowledge Graphs (KGs) are constructed to store structured facts about human knowledge or the objective world, and formalize facts as entities $e$ (nodes) and relations $r$ (links) between them. Static Knowledge Graphs (SKGs) and Temporal Knowledge Graphs (TKGs) are two typical forms of KGs. SKGs store facts in the form of triples $(e_s, r, e_o)$ and TKGs extend triples to quadruples $(e_s, r, e_o, t)$, where $t$ indicates the happening time. Since real-world events are usually ever-changing and associated with time, TKGs are naturally confronted with issues of continually emerging entities and facts in the future timestamps throughout their whole lifecycle (Chen et al., 2023a). Therefore, this paper investigates TKG link prediction task over incomplete TKGs in the lifelong setting, named lifelong

---
*Corresponding Author

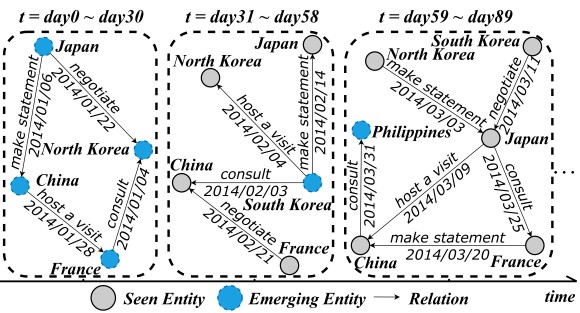

Figure 1: A sample sequence of three TKG snapshots in lifelong TKG reasoning. New entities (blue nodes) and new facts (all edges) in each snapshot emerge with time.

TKG reasoning. Figure 1 is an example in dataset ICEWS14 for temporally growing TKGs.

However, SKG reasoning methods (Trouillon et al., 2016) lack the consideration of temporal changing; conventional transductive TKG reasoning methods (Lacroix et al., 2020) need re-train for their closed-world assumption; and the latest inductive TKG reasoning methods (Chen et al., 2023b) treat emerging entities as simultaneous, oversimplifying the real scenario and thereby leading their genuine performance to be questionable. Hence, our proposed lifelong TKG reasoning issue is more challenging and realistic.

We formulate the lifelong TKG reasoning as a temporal-path-based RL framework and design the whole pipeline for extrapolating, transferring and updating. In the following, we introduce our targeted solutions and expound on their advantages over existing methods.

First, we focus on the temporal displacement between timestamps of candidate edge and its preceding edge and add temporal displacement into RL action space. TKGE models (Xu et al., 2021) rely on embeddings of absolute timestamps and are only fit for the past timestamps. Obviously, they do not meet the requirements of lifelong learning. On the contrary, our used transferable temporal dis-

placement in RL can be extrapolated from known timestamps to arbitrary future timestamps based on the magnitude of the displacement. In addition, we further design a reward-shaping module based on temporal rules found by RL, which only have the temporal order constraints of relations. This module makes the reasoning get rid of particular entities and will still be applicable for future timestamps.

Secondly, lifelong TKG reasoning can be considered as multiple consecutive inductive TKG reasoning processes. Recently, inductive TKG reasoning methods (Park et al., 2022; Xu et al., 2023) can only deal with future time, not new components, not to mention their ability to continuously learn as required in lifelong TKG reasoning. Therefore, we design a new edge-aware message passing module, which not only transfers learned relation types to initialize emerging entities, but also updates the embeddings of all entities and edges in new TKGs. We also use the embeddings of specific edges rather than immobile relation types, since we seek to explore the concrete environment for each fact to counteract the influence of rapid TKG growth.

We build three new benchmarks based on three popular datasets to simulate the lifelong TKG reasoning scenario. In the experiments, we carefully adapt existing baselines by empowering them with temporal extrapolation or knowledge transfer capabilities. In summary, our main contributions are:

- To our knowledge, we are the first to pose and explore the more challenging and realistic lifelong TKG reasoning issue, which simulates growing TKGs in terms of timestamps, entities and facts, and we formulate it as a RL-based framework.

- To solve the challenges of temporal extrapolation and knowledge transfer in lifelong TKG reasoning, we propose the targeted solutions: temporal displacement, temporal-rule-based reward shaping and an edge-aware message passing module.

- We build three new benchmarks to evaluate our model. Experiments on temporal link prediction show that our model not only achieves the best average performance but also has progressively improving results on growing TKG snapshots.

## 2 Related Works

### 2.1 Inductive SKG Reasoning

Traditional SKG reasoning models, such as SKG Embedding (SKGE) methods, focus on the trans-

ductive setting where they are trained and tested in a fixed set of components. Recently, inductive SKG reasoning has drawn much attention. GraIL (Teru et al., 2020), SE-GNN (Li et al., 2021a) and MaKEr (Chen et al., 2022) are all GNN-based inductive reasoning models, from the points of view of subgraphs, data relevance and meta-learning. PathCon (Wang et al., 2021) leverages relational message passing for relation prediction, however, we aim at the harder entity prediction. Moreover, MINERVA (Das et al., 2018) first introduces RL to search for the tail entity of each query end-to-end. Multi-Hop (Lin et al., 2018) advances MINERVA and does reward shaping based on SKGE methods.

### 2.2 Inductive TKG Reasoning

Inductive TKG reasoning models mainly deal with seen entities in the future time. xERTE (Han et al., 2021) is delicately designed to forecast future links by an iterative sampling of temporal neighbours. TGAP (Jung et al., 2021) introduces temporally relevant events in GNN for better explainability. For RL-based TKG reasoning, CluSTeR (Li et al., 2021b) regards RL as a clue searching stage, but it strips temporal dimension away from RL and then rearranges the clues in chronological order at the next temporal reasoning stage. However, they can not handle unseen entities emerging with time. TITer (Sun et al., 2021) further defines a relative time encoding to distinguish the same entity in different timestamps and leverage the query information to represent unseen entities. Different from the above models, we introduce temporal displacement of facts in the RL and propose relation-type-based knowledge transferring for emerging entities.

### 2.3 Lifelong KG Reasoning

Recently, how to retain and reuse previous knowledge in a new environment has become a research highlight (Wang et al., 2019). MBE (Cui et al., 2022) explores inductive SKG reasoning under the multi-batch emergence scenario, which is similar to the concept of lifelong KG reasoning without considering time. Next, LKGE (Cui et al., 2023) first formally studies lifelong SKG reasoning via transferring knowledge and using TransE (Bordes et al., 2013) as the base model. However, they did not pay attention to the crucial temporal factor in TKGs. To this end, we raise lifelong TKG reasoning, which involves both unseen components and future timestamps, making this issue challenging

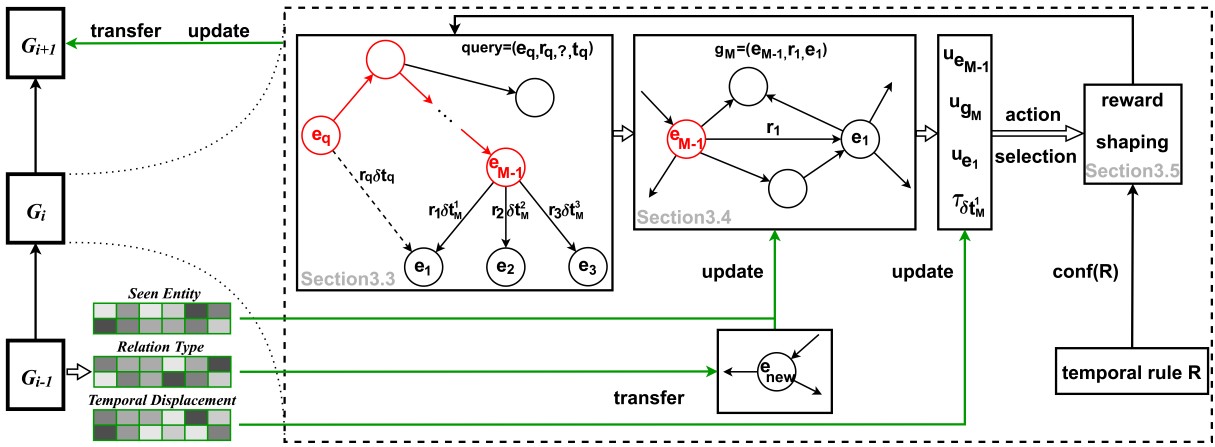

Figure 2: Model overview. $e_i$ is candidate entity and $\delta t_M^i$ is the corresponding temporal displacement ($i = 1, 2, 3$).

and realistic.

## 3 Methodology

### 3.1 Preliminaries

Growing TKGs in lifelong TKG reasoning can be viewed as a sequence of $\rho$ snapshots: $\mathcal{G} = (\mathcal{G}_1, \mathcal{G}_2, \ldots, \mathcal{G}_\rho)$, each of which, $\mathcal{G}_i$, contains a collection of fact quadruples over a continuous time period and $\mathcal{G}_i = \{\mathcal{E}_i, \mathcal{R}_i, \mathcal{T}_i, \mathcal{D}_i\}$. $\mathcal{E}_i, \mathcal{R}_i, \mathcal{T}_i, \mathcal{D}_i$ are entity, relation, timestamp and fact sets, and $\mathcal{E}_i \subsetneq \mathcal{E}_{i+1}$, $\mathcal{R}_i = \mathcal{R}_{i+1}, t_i < t_{i+1}$ ($t_i \in \mathcal{T}_i, t_{i+1} \in \mathcal{T}_{i+1}$), and $\mathcal{D}_i \cap \mathcal{D}_{i+1} = \emptyset$. We use $\mathcal{E}_{\Delta i+1} = \mathcal{E}_{i+1} - \mathcal{E}_i$ and $\mathcal{D}_{i+1}$ to denote the emerging entities and facts. The TKG link prediction task asks to predict the missing entities of the query edge in incomplete TKGs. For lifelong TKG reasoning, we leverage the TKG link prediction task above to train a new model $\mathcal{M}_{i+1}$ by transferring and updating knowledge in $\mathcal{M}_i$ to fit $\mathcal{E}_{\Delta i+1}$ and $\mathcal{D}_{i+1}$. Specifically, $\mathcal{D}_i$ in $\mathcal{G}_i$ is divided into a training set $\mathcal{F}_i$, a validation set $\mathcal{V}_i$ and a test set $\mathcal{Q}_i$. After finishing the training on $\mathcal{F}_i$ and validation on $\mathcal{V}_i$, the model $\mathcal{M}_i$ is evaluated on the accumulated test sets $\cup_{j=1}^i \mathcal{Q}_j$ for overall assessments.

### 3.2 Model Overview

Figure 2 is the architecture of our model. Along a sequence of growing TKGs, our model transfers and updates knowledge iteratively from the previous TKG snapshot to the next one, avoiding re-training. Inside each $\mathcal{G}_i$, we regard the reasoning as walk-based action selecting process. An agent starts from the query entity, constantly takes actions through temporal edges based on temporal displacement, and expects to reach the target entity within a limited number of steps (Section 3.3). To

achieve knowledge transferring and updating, we inject embeddings of relation types into emerging entities $e_{new}$, and then update all the embeddings of entities $e$ and edges $g$ in our proposed edge-aware message passing module (Section 3.4). Section 3.5 describes our designed temporal-rule-based reward shaping.

### 3.3 Reinforcement Learning Framework

For each edge, we add its reversed edge to $\mathcal{G}_i$, making the reasoning traceable and controllable. For each entity $e$, we also add self-loop temporal edges at every timestamp to $\mathcal{G}_i$, which allows the agent to stay in a place, and they work as stop actions.

#### 3.3.1 Environment Setting

Our environment can be formulated as a Markov Decision Process (MDP) over TKGs and has the following components. We take $\mathcal{G}_i$ as an example.

**States.** Let $\mathcal{S}_i$ and $(e_q, r_q, ?, t_q)$ denote all possible states of $\mathcal{G}_i$ and the query. At step $m \in [0, M]$, the agent locates at entity $e_m$ and timestamp $t_m$, so the state $s_m = (e_m, t_m, e_q, r_q, t_q) \in \mathcal{S}_i$. Specifically, the initial state is $s_0 = (e_q, t_q, e_q, r_q, t_q)$.

**Time-constrained Actions.** Let $\mathcal{A}_i$ denote the action space of $\mathcal{G}_i$. Let $\mathcal{A}_i^m$ denote the set of optional actions of $s_m$ in $\mathcal{G}_i$. Compared with SKGs, the time dimension causes the action space of RL in TKGs extremely large. Hence, we add two temporal constraints to prune the action space, since facts closer to $t_m$ in the state $s_m$ under consideration are more likely to be directly relevant to the prediction:

$$\mathcal{A}_i^m = \{(e', g', t_m - t') \mid g' = (e_m, r', e', t') \in \mathcal{G}_i, \\ t_m - t' \leq T, \quad t' \leq t_m \leq t_q\}, \tag{1}$$

where $g'$ is a candidate edge, $t_m - t'$ is the temporal displacement, $T$ is a hyperparameter. $\mathcal{A}_i^m$ naturally considers reversed and self-loop temporal edges.

**Transitions.** The transition function $\omega : \mathcal{S}_i \times \mathcal{A}_i \to \mathcal{S}_i$ is deterministic under $\mathcal{G}_i$ and updates the states depending on the selected actions.

**Rewards.** In the default formulation, agents receive reward $R_b(s_M) = \mathbb{I}(e_M == e_{ans})$, where $s_M = (e_M, t_M, e_q, r_q, t_q)$ is the final state, $e_{ans}$ is the answer to the query and $\mathbb{I}$ is a binary indicator function. Our designed reward $R(s_M)$ shaped by temporal rules will be described in Section 3.5.

### 3.3.2 Policy Network

First, the temporal displacement between timestamps of current state $t_{m-1}$ ($t_0 = t_q$) and its subsequent action $t_m$ can integrally capture the time-related dynamics. The temporal displacement is donated as $\delta t_m = t_{m-1} - t_m \leq T$. Secondly, because of temporally evolving TKGs, even if the relation types of two edges in $\mathcal{G}_i$ and $\mathcal{G}_{i+1}$ are the same, their semantics can change considerably due to different surrounding environments. It is only by taking surrounding edges into account that we can understand their contextual semantics. Moreover, the above operations are also in line with the foundation of RL, which is constant interaction with the environment.

Therefore, the input of policy network has three parts: $\mathbf{u}_{e_m}, \mathbf{u}_{g_m}, \boldsymbol{\tau}_{\delta t_m} \in \mathbb{R}^d$, i.e., the embeddings of entity $e_m$, edge $g_m = (e_{m-1}, r_m, e_m)$, and temporal displacement $\delta t_m$ ($\mathbf{u}_{e_m}$, $\mathbf{u}_{g_m}$ are obtained from edge-aware message passing module in Section 3.4, $\boldsymbol{\tau}_{\delta t_m}$ is transferred from $\mathcal{G}_{i-1}$ and updated in $\mathcal{G}_i$). Path history $\mathbf{h}_m$ in $\mathcal{G}_i$ is encoded as follows:

$$
\begin{aligned}
\mathbf{h}_m &= \text{LSTM}\left(\mathbf{h}_{m-1}, [\mathbf{u}_{e_m}, \mathbf{u}_{g_m}, \boldsymbol{\tau}_{\delta t_m}]\right); \\
\mathbf{h}_0 &= \text{LSTM}\left(\mathbf{0}, [\mathbf{u}_{e_q}, \mathbf{u}_{r_0}, \boldsymbol{\tau}_{\delta t_q}]\right),
\end{aligned}
\tag{2}
$$

where $\mathbf{h}_m \in \mathbb{R}^{2d}$, $\mathbf{u}_{r_0} \in \mathbb{R}^d$ is the embedding of the special starting relation $r_0$, and $\delta t_q = 0$. For a candidate next action $a' = (e', g', \delta t') \in \mathcal{A}_i^m$ ($\delta t' = t_m - t'$) in Eq. 1, we calculate the probability of its state transition based on the correlation of the action and the query in terms of entities and edges:

$$
\begin{aligned}
\phi\left(a', s_m\right) = {} & \lambda \langle [\mathbf{u}_{e'}, \boldsymbol{\tau}_{\delta t'}], \mathbf{W}_e \mathbf{E}_q \rangle \\
& + (1 - \lambda) \langle [\mathbf{u}_{g'}, \boldsymbol{\tau}_{\delta t'}], \mathbf{W}_g \mathbf{E}_q \rangle;
\end{aligned}
$$

$$
\begin{aligned}
\mathbf{E}_q &= \text{ReLU}\left(\mathbf{W}_q\left[\mathbf{u}_{e_q}, \mathbf{u}_{r_q}, \boldsymbol{\tau}_{\delta t_q}, \mathbf{h}_m\right]\right); \\
\lambda &= \sigma\left(\mathbf{W}_\lambda\left[\mathbf{u}_{e'}, \mathbf{u}_{g'}, \boldsymbol{\tau}_{\delta t'}, \mathbf{u}_{e_q}, \mathbf{u}_{r_q}, \boldsymbol{\tau}_{\delta t_q}, \mathbf{h}_m\right]\right),
\end{aligned}
\tag{3}
$$

where $\mathbf{W}_e, \mathbf{W}_g, \mathbf{W}_q, \mathbf{W}_\lambda$ are learnable matrices, $\langle \cdot, \cdot \rangle$ is vector dot product. After scoring all actions, policy network $\pi_\theta(a_{m+1}|s_m)$ is obtained through softmax.

### 3.3.3 Training and Optimization

We fix the search path length to $M$. In lifelong TKG reasoning, the policy network is trained by maximizing the expected reward over growing TKG snapshots $\mathcal{G}_1, \mathcal{G}_2, \ldots, \mathcal{G}_\rho$. Hence, our model is required to train over $\mathcal{F}_1, \mathcal{F}_2, \ldots, \mathcal{F}_\rho$ in turn:

$$
\begin{aligned}
J(\theta) = \mathbb{E}_{(e_q, r_q, e_{ans}, t_q) \sim \mathcal{F}_i}[\mathbb{E}_{a_1, \ldots, a_M \sim \pi_\theta} \\
[R(s_M \,|\, e_q, r_q, t_q)]],
\end{aligned}
\tag{4}
$$

where $i \in [1, \rho]$. Then, we use the REINFORCE algorithm to iteratively optimize our model:

$$
\nabla_\theta J(\theta) \approx \nabla_\theta \sum_{m \in [0, M]} R(s_M \,|\, e_q, r_q, t_q) \log \pi_\theta.
\tag{5}
$$

### 3.4 Embedding Transfer and Update

SKGs with static entity properties can be seen as long-term valid knowledge and are helpful to generate accurate evolutional embeddings of entities (Li et al., 2021c; Niu and Li, 2023). Therefore, for each TKG snapshot $\mathcal{G}_i$, the timestamps are masked to convert $\mathcal{G}_i$ to its corresponding SKG snapshot $\mathcal{G}_i^s$.

We adopt a relation-type-based transferring layer over $\mathcal{G}_i^s$, since the relation types of connected edges provide valuable clues about their natures. Our introduced transferring layer injects learned knowledge about relation types into new entities. Formally, for a new entity $e$ in $\mathcal{G}_i$, we generate its beginning embedding $\mathbf{u}_i^b(e) \in \mathbb{R}^d$:

$$
\begin{aligned}
\mathbf{u}_i^b(e) = \tanh\Big( & \sum_{(e', r, e) \in \mathcal{N}_i^{\text{in}}(e)} \mathbf{W}_{\text{in}}\, \mathbf{u}_r \\
& + \sum_{(e, r, e') \in \mathcal{N}_i^{\text{out}}(e)} \mathbf{W}_{\text{out}}\, \mathbf{u}_r \Big),
\end{aligned}
\tag{6}
$$

where $\mathcal{N}_i^{\text{in}}(e) = \{(e', r, e) \,|\, (e', r, e) \in \mathcal{G}_i^s\}$, $\mathcal{N}_i^{\text{out}}(e) = \{(e, r, e') \,|\, (e, r, e') \in \mathcal{G}_i^s\}$. $\mathbf{W}_{\text{in}}, \mathbf{W}_{\text{out}} \in \mathbb{R}^{d \times d}$ are two learnable weight matrices. $\mathbf{u}_r \in \mathbb{R}^d$, serving as the embedding of relation type $r$, is learnable throughout the whole lifelong TKG reasoning process. Furthermore, to avoid recalculating for seen entities, we only generate embeddings for emerging entities and inherit the

embeddings for seen ones from the preceding TKG snapshot $\mathcal{G}_{i-1}$.

In order to update embeddings for all components in $\mathcal{G}_i^s$, we propose a new edge-aware message passing module via bidirectional communication between edges and nodes. This module enables our model to better adapt to the rapidly changing environment in lifelong TKG learning. For each edge $g$ in $\mathcal{G}_i^s$, the links connected to its two endpoints act as a relevant semantic environment.

Therefore, we alternately pass edge-aware messages between nodes and edges to aggregate unique environment knowledge for each edge as follows. $\mathbf{u}_i^\ell(e)$ and $\mathbf{u}_i^\ell(g)$ are embeddings of entity $e$ and edge $g$ at $\ell$-th layer:

$$\mathbf{u}_i^{\ell+1}(e) = \tanh\Big(\mathbf{W}_{\text{self}}^\ell \mathbf{u}_i^\ell(e)$$
$$+\sum\nolimits_{g'=(e',r,e)\in\mathcal{N}_i^{\text{in}}(e)} \mathbf{W}_{\text{in}}^\ell \varphi\Big(\mathbf{u}_i^\ell(e'), \mathbf{u}_i^\ell(g')\Big)$$
$$+\sum\nolimits_{g'=(e,r,e')\in\mathcal{N}_i^{\text{out}}(e)} \mathbf{W}_{\text{out}}^\ell \varphi\Big(\mathbf{u}_i^\ell(e'), \mathbf{u}_i^\ell(g')\Big)\Big);$$

$$\qquad\qquad\qquad\qquad\qquad\qquad (7)$$

$$\mathbf{u}_i^{\ell+1}(g) =$$
$$\sigma\Big(\mathbf{W}_g^\ell\Big[\mathbf{u}_i^\ell(g), \mathbf{u}_i^{\ell+1}(e_{left}), \mathbf{u}_i^{\ell+1}(e_{right})\Big]+\mathbf{b}_g^\ell\Big),$$

$$\qquad\qquad\qquad\qquad\qquad\qquad (8)$$

where $\mathbf{u}_i^0(e) = \mathbf{u}_i^b(e)$, $\mathbf{u}_i^0(e') = \mathbf{u}_i^b(e')$, $\mathbf{u}_i^0(g) = \mathbf{u}_r$, $\mathbf{u}_i^0(g') = \mathbf{u}_{r'}$, $r(r')$ is the relation type of $g(g')$, $e_{left}$ and $e_{right}$ are two endpoints of edge $g$. $\mathbf{W}_{\text{self}}^\ell$, $\mathbf{W}_{\text{in}}^\ell$, $\mathbf{W}_{\text{out}}^\ell \in \mathbb{R}^{d\times d}$; $\mathbf{W}_g^\ell \in \mathbb{R}^{d\times 3d}$ and $\mathbf{b}_g^\ell \in \mathbb{R}^d$ are learnable weight matrices. Message transition function $\varphi(\mathbf{u}_i^\ell(e'), \mathbf{u}_i^\ell(g'))=\mathbf{u}_i^\ell(e') \circ \mathbf{u}_i^\ell(g')$ stores environment knowledge by calculating the correlation between connected entities and edges. $\circ$ is hadamard product.

After L-layer updating, the final representations of each entity $e$ and edge $g$ are $\mathbf{u}_i^L(e)$ and $\mathbf{u}_i^L(g)$. In the absence of ambiguity, we abbreviate them in RL (Section 3.3) as $\mathbf{u}_e$ and $\mathbf{u}_g$, respectively.

### 3.5 Temporal-Rule-Based Reward Shaping

For a query $(e_q, r_q, ?, t_q)$ with answer $e_{ans}$, RL gives a reasoning trajectory $((e_q, r_1, e_1, t_1), (e_1, r_2, e_2, t_2), \ldots, (e_{M-1}, r_M, e_M, t_M))$, where $t_q \geq t_1 \geq t_2 \geq \cdots \geq t_M$. Then, we can extract a temporal rule $R : (r_M, \ldots, r_2, r_1) \Rightarrow r_q$ with non-descending temporal constraints and denote the confidence of $R$ as $conf(R)$. According to Section 3.3.1, since the agent receives a binary reward only based on whether $e_M$ is equal to $e_{ans}$, regardless of the quality of the reasoning temporal paths,

we introduce a temporal-rule-based reward shaping to guide the training of the agent:

$$R(s_M) = R_b(s_M) + conf(R). \qquad (9)$$

$conf(R)$ is obtained by dividing the *rule support* by the *body support*.

## 4 Constructed Benchmarks

To conduct evaluation for lifelong TKG reasoning, we construct three new TKG benchmarks based on three datasets ICEWS14, ICEWS05-15 (García-Durán et al., 2018) and ICEWS18 (Jin et al., 2020) to simulate their growth situation of entities, named as ICEWS14-lifelong, ICEWS05-15-lifelong and ICEWS18-lifelong. Table 1 shows the statistics of the new benchmarks.

1. **Counting.** More uniform entity distribution makes the changing of $\mathcal{G}$ more concentrated and significant, so we filter entities that occur less than 10 times and count the remaining entities, relations, timestamps as $|\mathcal{E}|, |\mathcal{R}|, |\mathcal{T}|$.

2. **Lifelong Simulating.** New entities emerge at almost all the timestamps. Hence, we accumulate the number of entities in chronological order. First, we define the set of entities at $t_0 = 0$ as the initial $\mathcal{E}_i$ ($i = 1, \ldots, 5$). Secondly, we iteratively add the set of entities at the next timestamp to expand $\mathcal{E}_i$. Once $|\mathcal{E}_i| \geq \frac{4+i}{10}|\mathcal{E}|$, we stop expanding $\mathcal{E}_i$, record the current timestamp $t_i$ and then start the searching for $t_{i+1}$ in the same way. Thirdly, after obtaining five timestamps $t_1 \sim t_5$, we denote the union of TKGs from $t_{i-1}$ to $t_i$ as TKG snapshot $\mathcal{G}_i$. Since the relations are dense, we can ensure the number of relations in all $\mathcal{G}_i$, $|\mathcal{R}_i|$, is equal to $|\mathcal{R}|$. For the last TKG snapshot $\mathcal{G}_6$, we set $t_6 = |\mathcal{T}|$ to ensure all facts in $\mathcal{D}$ are covered.

3. **Dividing.** For each TKG snapshot $\mathcal{G}_i$, we randomly divide $\mathcal{G}_i$ into training set $\mathcal{F}_i$, validation set $\mathcal{V}_i$ and test set $\mathcal{Q}_i$ with ratio 3:1:1.

| Benchmarks | $\mathcal{G}_1$ | | | $\mathcal{G}_2$ | | | $\mathcal{G}_3$ | | | $\mathcal{G}_4$ | | | $\mathcal{G}_5$ | | | $\mathcal{G}_6$ | | |
|---|---|---|---|---|---|---|---|---|---|---|---|---|---|---|---|---|---|---|
| | $|\mathcal{E}_1|$ | $|\mathcal{D}_1|$ | $|\mathcal{T}_1|$ | $|\mathcal{E}_2|$ | $|\mathcal{D}_2|$ | $|\mathcal{T}_2|$ | $|\mathcal{E}_3|$ | $|\mathcal{D}_3|$ | $|\mathcal{T}_3|$ | $|\mathcal{E}_4|$ | $|\mathcal{D}_4|$ | $|\mathcal{T}_4|$ | $|\mathcal{E}_5|$ | $|\mathcal{D}_5|$ | $|\mathcal{T}_5|$ | $|\mathcal{E}_6|$ | $|\mathcal{D}_6|$ | $|\mathcal{T}_6|$ |
| ICEWS14-lifelong | 3,609 | 19,690 | 83 | 4,333 | 9,282 | 39 | 5,075 | 11,328 | 45 | 5,803 | 14,937 | 62 | 6,533 | 17,531 | 67 | 7,128 | 17,962 | 69 |
| ICEWS05-15-lifelong | 1,065 | 9,197 | 457 | 1,157 | 3,843 | 172 | 1,249 | 5,296 | 229 | 1,342 | 8,255 | 394 | 1,434 | 19,064 | 917 | 1,520 | 33,201 | 1,848 |
| ICEWS18-lifelong | 1,782 | 5,418 | 29 | 1,950 | 1,601 | 8 | 2,107 | 2,661 | 13 | 2,265 | 3,854 | 18 | 2,422 | 8,832 | 39 | 2,519 | 38,207 | 197 |

Table 1: Statistics of the constructed benchmarks. $|\mathcal{E}_i|$, $|\mathcal{D}_i|$, $|\mathcal{T}_i|$ are numbers of entities, facts and timestamps in $\mathcal{G}_i$.

# 5 Experiments

## 5.1 Experimental Setup

**Baselines.**

We compare our model against the SOTA lifelong SKG reasoning model LKGE (Cui et al., 2023). LKGE uses TransE as the knowledge transfer module and can be well-adapted to lifelong TKG reasoning. MBE (Cui et al., 2022) is designed for multi-batch emergence scenario and is also a powerful baseline since it leverages walk-based reasoning and has an inductive entity encoding module. We name MBE adapted for lifelong TKG reasoning as L-MBE.

Based on the framework of LKGE, we leverage the SOTA TKGE model TeRo (Xu et al., 2020) for L-TeRo, by defining the timestamp embeddings as a linear function. L-TITer, L-TGAP and L-TLogic were modified from TITer (Sun et al., 2021), TGAP (Jung et al., 2021) and TLogic (Liu et al., 2022). TITer is encapsulated according to the requirements of lifelong TKG reasoning, forming L-TITer. TGAP can handle future timestamps, but it is not inductive, so L-TGAP randomly initializes the emerging entities. TLogic is based on entity-independent temporal logical rules, so L-TLogic can be used for lifelong TKG reasoning by transferring its found rules.

**Evaluation Metrics.**

Following the convention, we conduct the experiments on the temporal link prediction task and report Mean Reciprocal Rank (MRR) and Hits@$k$ ($k = 3, 10$). Then, we evaluate the knowledge lifelong learning capability using forward transfer (FWT) and backward transfer (BWT) (Lopez-Paz and Ranzato, 2017).

$$FWT = \frac{1}{\rho - 1}\sum_{i=2}^{\rho} h_{i-1,i}; BWT = \frac{1}{\rho - 1}\sum_{i=1}^{\rho-1}(h_{\rho,i} - h_{i,i}),$$

where $\rho$ is the number of TKG snapshots, $h_{i,j}$ is the MRR scores on $\mathcal{Q}_j$ after training the model $\mathcal{M}_i$ on $\mathcal{F}_i$.

## 5.2 Results on Lifelong TKG Reasoning

We run 10-seed experiments for all models on our three new benchmarks and report average results on the six TKG snapshots. The MRR, Hits@3 and Hits@10 results are shown in Table 2. Although the TransE used in LKGE is efficient to model static relationships of entities, it is not sensitive to temporal change. Therefore, LKGE does not deal with the temporal dimension in lifelong TKG reasoning and confuses the timestamps in growing TKGs. As a result, LKGE only has comparable Hits@10 results. Similar to our model, the adapted L-MBE uses a RL framework, benefiting its reasoning process. The static link augmentation in L-MBE is also proven to be effective in TKG reasoning (Niu and Li, 2023). Even so, the results of L-MBE are still worse than ours. The reason is that our model specifically constructs a more targeted RL environment for lifelong TKG reasoning and further takes temporal displacement into account for temporally extrapolating.

However, the frameworks of LKGE and L-TeRo are based on traditional KGE and TKGE models, limiting their performances in lifelong TKG reasoning. L-TITer, L-TGAP and L-TLogic are adapted from recent powerful TKG link prediction models, but there is a certain degree of reduction in their original performances when used in lifelong TKG reasoning. The reasons are that the IM module in TITer is parameter-free, so the knowledge can not be transferred and updated; TGAP is in the interpolation setting, so the embeddings of future timestamps can not be well initialized; TLogic relies on temporal rules, but the transferred old rules may conflict with the new ones in future TKG snapshots. Therefore, these existing baselines can not perform perfectly in the lifelong TKG reasoning. On the contrary, our model consistently outperforms all the baselines across the three benchmarks. Some results of our model are even twice that of baselines. This is because our model comprehensively considers all the requirements of lifelong TKG reasoning and the proposed three targeted solutions solve the problems of temporal extrapolation, knowledge transfer and knowledge update.

|  | ICEWS14-lifelong | | | ICEWS05-15-lifelong | | | ICEWS18-lifelong | | |
|---|---|---|---|---|---|---|---|---|---|
|  | MRR | Hits@3 | Hits@10 | MRR | Hits@3 | Hits@10 | MRR | Hits@3 | Hits@10 |
| LKGE | 0.178 | 22.73 | 44.31 | 0.214 | 34.91 | 53.73 | 0.132 | 17.78 | 31.77 |
| L-MBE | 0.349 | 37.32 | 46.93 | 0.287 | 32.39 | 44.80 | 0.155 | 16.97 | 25.03 |
| L-TeRo | 0.182 | 27.34 | 47.03 | 0.280 | 36.95 | 55.20 | 0.196 | 26.14 | 34.94 |
| L-TITer | 0.193 | 21.95 | 33.26 | 0.275 | 31.87 | 48.25 | 0.249 | 31.42 | 40.35 |
| L-TGAP | 0.220 | 24.84 | 36.92 | 0.309 | 34.71 | 51.45 | 0.293 | 37.52 | 43.67 |
| L-TLogic | 0.323 | 36.83 | 47.28 | 0.296 | 33.73 | 45.67 | 0.251 | 37.40 | 39.09 |
| Ours | **0.377** | **39.64** | **49.79** | **0.323** | **38.25** | **57.35** | **0.385** | **46.56** | **55.26** |

Table 2: Average lifelong TKG reasoning results on the six TKG snapshots (% for Hits@3 and Hits@10). **Bold** numbers denote the best results.

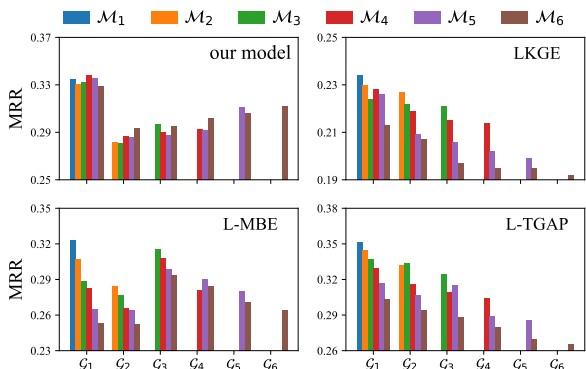

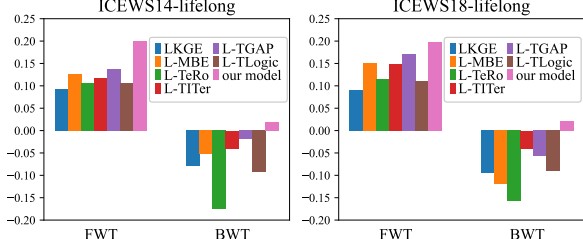

Figure 4: FWT and BWT of MRR results on two benchmarks: ICEWS14-lifelong and ICEWS18-lifelong.

Figure 3: MRR changes on ICEWS05-15-lifelong of our model and three baselines: LKGE, L-MBE, L-TGAP.

## 5.3 Performance Evolution

To demonstrate the performance evolution of our model and three baselines during lifelong TKG reasoning, the MRR results on all TKG snapshots are reported in Figure 3.

We find:

(i) For the same $\mathcal{G}_i$, as our model learns over growing TKGs, the performance of $\mathcal{M}_i \sim \mathcal{M}_6$ on $\mathcal{G}_i$ remains relatively steady. This suggests that our model has a strong ability to transfer embeddings to emerging components continuously and avoids forgetting previous knowledge. However, the other three baselines all experience rapid degradation in performance and suffer from catastrophic forgetting.

(ii) For the same $\mathcal{M}_i$, we observe its performance changing on different TKG snapshots. Since old knowledge is not always suitable for new facts, we need to update learned knowledge, otherwise the performance of $\mathcal{M}_i$ will drop. The inductive embedding layer in L-MBE sometimes succeeds in updating (from $\mathcal{G}_2$ to $\mathcal{G}_3$). LKGE and L-TGAP clearly fail to update knowledge for the decreasing MRR. This demonstrates our powerful ability to

update knowledge for new TKGs. Empirically, performance on $\mathcal{G}_1$ is the upper bound of our model since $\mathcal{G}_1$ injects knowledge to our model and knowledge transfer and update start when $\mathcal{G}_1$ grows to $\mathcal{G}_2$.

## 5.4 Knowledge Lifelong Learning Capability

To accurately quantify the knowledge lifelong learning capability of all models in lifelong TKG reasoning, we report the FWT and BWT scores of MRR results in Figure 4. Since we specifically design the whole reasoning pipeline for extrapolating and transferring, the FWT scores of our model are the best among all baselines. LKGE and L-MBE are originally suitable for lifelong KG reasoning, so they work relatively well over SKGs. The FWT scores of L-TLogic are also poor because its rule-based strategy is too restricted by symbolic relations to reason over future TKGs.

The BWT scores of baselines are all negative due to the overwriting of relation types embeddings as the result of their update of learned knowledge. On the contrary, our proposed edge-aware message passing module focuses on updating the unique environment of each fact rather than individual relation types, which makes our model introduce richer information for more accurate predictions. The low BWT scores of L-TeRo show the inefficiency of

| Query : (Sudan, sign formal agreement, Ethiopia, 2014/12/25) | Confidence |
|---|---|
| (Sudan, praise or endorse, Ethiopia, 2014/11/26)
→ (Ethiopia, provide military aid, Sudan, 2014/10/19)
→ (Sudan, cooperate economically, Ethiopia, 2014/09/15) | 0.75 |
| (Sudan, sign formal agreement, South Sudan, 2014/11/26)
→ (South Sudan, express to meet, Ethiopia, 2014/10/22) | 0.64 |
| (Sudan, Criticize or denounce, Barack Obama, 2014/12/12)
→ (Barack Obama, sign formal agreement, China, 2014/12/01) | 0.29 |

Table 3: Reasoning trajectories and confidence of corresponding temporal rules. Target entities are underlined.

| Benchmarks | Ours | w/o td | w/o mp | w/o rs |
|---|---|---|---|---|
| ICEWS14-lifelong | **0.377** | 0.321 | 0.335 | 0.349 |
| ICEWS05-15-lifelong | **0.323** | 0.268 | 0.284 | 0.291 |
| ICEWS18-lifelong | **0.385** | 0.299 | 0.349 | 0.362 |

Table 4: MRR results of ablation study for our model on the three new benchmarks.

TKGE in lifelong TKG reasoning. L-TITer and L-TGAP are well-adapted baselines by combining their original TKG reasoning ability with the demands of lifelong TKG reasoning, so their BWT scores are better than other baselines.

### 5.5 Case Study

**Reasoning Trajectories and Temporal Rules.**
To demonstrate the reasoning ability of our model over the temporal edges, we perform a case study in Table 3, which shows two positive reasoning trajectories and a negative one for the target query (*Sudan*, *sign formal agreement*, *Ethiopia*, *2014/12/25*). Then we further give the scores of confidence of the corresponding temporal rules extracted from the reasoning trajectories. The length of paths is set to 3 and we remove the self-loop actions for clarity. It can be seen that our model succeeds in telling the reasonable temporal rules with high confidence from weak ones with low confidence and thereby guides the action selection stably and efficiently.

### 5.6 Ablation Study

To further examine the effect of the three proposed solutions for lifelong TKG reasoning in our model, we conduct an ablation study as shown in Table 4.

First, we replace temporal displacement in RL with timestamps (w/o td), i.e., rely on embeddings of explicit timestamps while selecting actions. The

MRR results decrease by 18.07% on average over three benchmarks, indicating the significance of considering temporal displacement in RL, because it can be temporally extrapolated to the future time.

Secondly, we remove our proposed edge-aware message passing module (w/o mp) and randomly initialize new entities. In this case, our model can not transfer or update knowledge and the agent can not capture the specific environment of each edge in RL. This leads to a drop of 10.86% on average on MRR, implying the importance of this module.

Finally, we leverage the original binary reward to replace the temporal-rule-based reward shaping in our model (w/o rs) and obverse a 7.77% performance degradation. This phenomenon means, by the training guidance of temporal rules independent of particular entities, our model obtains comprehensively enhanced ability to select reliable actions.

### 6 Conclusion

In this work, the lifelong TKG reasoning issue involves continually emerging entities and facts in the future timestamps. To address this new problem, we propose our model under the framework of RL and our model uses temporal displacement in the action space to extrapolate to the future timestamps; uses a new edge-aware message passing module to inductively transfer and update learned knowledge to new entities and facts; and uses a temporal-rule-based reward shaping to guide the training. The experimental results on three newly constructed benchmarks illustrate that our model has the best performance for lifelong TKG reasoning and the strongest knowledge lifelong learning capability.

## Limitations

This paper mainly focuses on lifelong TKG reasoning, where we consider emerging entities and facts in the future timestamps as TKG grows, and do not consider the changing case of relations. In most cases, the number of entities in TKGs is much larger than that of relations and the emergence of entities lasts longer and is more common than that of relations. For instance, ICEWS14 has 7128 entities and 230 relations; the accumulated number of relation types in ICEWS14 increases rapidly to 115, half of the total number, in 10 days, on the contrary, the accumulated number of entities is steadily increasing over the entire time period of ICEWS14. Therefore, we study the more severe case of emerging entities and leave the research for emerging relations in lifelong TKG reasoning to future works.

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

## A  Appendix

### A.1  Implementation Details.

we tune the hyperparameters of our model using grid-search: learning rate in {0.0005, 0.0001, 0.001}, batch size in {4096, 8192}, embedding dimension in {100, 200, 300}. The length of the walking path of the agent in RL, M, is tuned in {3, 4, 5}. The layer of our proposed edge-aware message passing module, L, is in a range of {1, 2, 3} The hyperparameter T in Section 3.3.1 is 40 in ICEWS14-lifelong, 300 in ICEWS05-15-lifelong, 29 in ICEWS18-lifelong. While training, The discount factor of REINFORCE is 0.95. We clip gradients greater than 20 to avoid the gradient explosion. While testing, we use beam search to obtain a list of predicted entities with the corresponding scores. The beam size is set to 100. The re-run baselines are based on their public codes and further adapted for our constructed benchmarks for lifelong TKG reasoning. All the experiments are carried out on one A100 GPU.

### A.2  Computational Complexity Analysis.

To see the efficiency of our proposed model, we analyze the computational complexity of the process of knowledge transferring and updating. There are two parts in the edge-aware message passing module, one is for initializing the embeddings for new entities, and the other is for updating all components. First, for the first part, the expected time complexity of Eq. 6 in each iteration is $\mathcal{O}(|\mathcal{E}|)$, where $|\mathcal{E}|$ is the maximum number of new entities in the six TKG snapshots. Secondly, for the next

part, if there are $X$ emerging entities and $Y$ edges connected to new entities, each emerging entity takes $\mathbb{E}[d] = \frac{2Y}{X}$ elements as input in expectation, where $\mathbb{E}[d]$ is the expected node degree. The cost of aggregation is $X \cdot \mathbb{E}[d] = 2Y$. Therefore, Eq. 7 has the time complexity $\mathcal{O}(2|\mathcal{D}|)$. For Eq. 8, it is performed for $|\mathcal{D}|$ times and each update takes 3 elements as input. Therefore, the cost of update in each iteration is $\mathcal{O}(3|\mathcal{D}|)$. We iterate these two updating functions for L times, so the overall time complexity is $\mathcal{O}(L|\mathcal{D}|)$, where $|\mathcal{D}|$ is the maximum number of edges in the six TKG snapshots. Finally, the time complexity of the process of knowledge transferring and updating is $\mathcal{O}(|\mathcal{E}| + L|\mathcal{D}|)$, where $|\mathcal{E}|$ is the maximum number of new entities, $|\mathcal{D}|$ is the maximum number of edges in the six TKG snapshots; L is the iteration time.