# OpenReview forum: "Temporal Extrapolation and Knowledge Transfer for Lifelong Temporal Knowledge Graph Reasoning"
_EMNLP/2023/Conference — EMNLP 2023 Findings_

### Official Review · Reviewer_kB2c · 2023-08-05

**Soundness:** 4

**Excitement:**

4: Strong: This paper deepens the understanding of some phenomenon or lowers the barriers to an existing research direction.

**Missing References:**

There are some missing references on temporal knowledge graph forecasting:

Trivedi, Rakshit, Hanjun Dai, Yichen Wang, and Le Song. "Know-evolve: Deep temporal reasoning for dynamic knowledge graphs." In international conference on machine learning, pp. 3462-3471. PMLR, 2017.

Han, Zhen, Zifeng Ding, Yunpu Ma, Yujia Gu, and Volker Tresp. "Learning neural ordinary equations for forecasting future links on temporal knowledge graphs." In Proceedings of the 2021 conference on empirical methods in natural language processing, pp. 8352-8364. 2021.

Besides, there are also some missing reference on learning unseen entities on TKG reasoning such as

Ding, Zifeng, Jingpei Wu, Bailan He, Yunpu Ma, Zhen Han, and Volker Tresp. "Few-Shot Inductive Learning on Temporal Knowledge Graphs using Concept-Aware Information." AKBC 2022.

**Paper Topic And Main Contributions:**

The paper studies the lifelong temporal knowledge graph reasoning problem, where a model is able to extrapolate missing links at future unseen timestamps and transfer learned graph embeddings to new emerging entities. The author formulate the lifelong TKG reasoning as a temporal-path-based RL framework. Specifically, the author focus on the temporal displacement between timestamps of candidate edge and its preceding edge and add temporal displacement into RL action space. Besides, the paper introduced a new edge-aware message passing module, which transfers learned relation types to initialize emerging entities. The author constructs three new datasets based on three benchmark datasets to simulate the lifelong TKG reasoning scenario. Experiments on temporal link prediction demonstrate the effectiveness of the proposed approach.

**Questions For The Authors:**

In Section 5.3, the author mentions that the proposed model has a strong ability to avoids forgetting previous knowledge. Are there any potential explanation besides the empirical results?

**Reasons To Accept:**

The paper is the first to explore the lifelong TKG reasoning problem, which simulates growing TKGs in terms of unseen timestamps, new emerging entities and events. The approach inject embeddings of relation types into emerging entities for knowledge transfer and update.

Besides, the paper proposed a reinforcement learning framework for solving the lifelong TKG reasoning task, where an agent starts from the query entity, constantly takes actions through temporal edges based on temporal displacement, and to search the target entity.

Moreover, the author also introduced three new TKG benchmarks for lifelong reasoning based on three datasets ICEWS14, ICEWS05-15, and ICEWS18, which could foster future research. In particular, the author simulate the reality by accumulating the number of entities in chronological order, and thus, let new entities emerge at almost all the timestamps. Empirical results on the datasets show the superiority of the proposed method compared to other baselines.

The author also investigated the performance evolution of the proposed model, demonstrating that the model’s performance remains relatively steady over growing temporal knowledge graphs.

**Reasons To Reject:**

The author only evaluates the model on ICEWS datasets but not on large-scale datasets such as GDELT, which contains several millions quadruples and more realistic.

**Reproducibility:**

4: Could mostly reproduce the results, but there may be some variation because of sample variance or minor variations in their interpretation of the protocol or method.

**Reviewer Confidence:**

4: Quite sure. I tried to check the important points carefully. It's unlikely, though conceivable, that I missed something that should affect my ratings.

---

> ### Author Rebuttal · Authors · 2023-08-29
>
> __Response to the Concerns of Reviewer kB2c.__
>
> We sincerely thank you for providing supportive comments and acknowledging the technical soundness of our work. We respond to your comments as follows.
>
> > Question 1.  The author only evaluates the model on ICEWS datasets but not on large-scale datasets such as GDELT, which contains several millions quadruples and more realistic.
>
> __Response to Question 1__:Thanks for your suggestions. Our experiments ask to reprocess the datasets and GDELT is very large. We plan to conduct the experiments for GDELT in the future.
>
>
>
> > Question 2.  In Section 5.3, the author mentions that the proposed model has a strong ability to avoid forgetting previous knowledge. Is there any potential explanation besides the empirical results?
>
> __Response to question 2__: This is a natural and good question.
>
> The learned knowledge can be inherited from the preceding TKG snapshot to the subsequent TKG snapshot.
> So, the previous knowledge is well remembered and utilized in our model to avoid forgetting.
>
> We will cite the missing papers.

---

### Official Review · Reviewer_dRYt · 2023-08-07

**Soundness:** 3

**Excitement:**

3: Ambivalent: It has merits (e.g., it reports state-of-the-art results, the idea is nice), but there are key weaknesses (e.g., it describes incremental work), and it can significantly benefit from another round of revision. However, I won't object to accepting it if my co-reviewers champion it.

**Paper Topic And Main Contributions:**

The paper first explored the lifelong TKG reasoning and proposed a model under the framework of RL. In the experiments, they illustrate that their model shows the best performance.

**Questions For The Authors:**

The authors provided the code for reproduction but I couldn't access the link.

**Reasons To Accept:**

1. This paper proposed a new task setup, the lifelong TKG reasoning, and construct datasets for the task.
2. The proposed method outperforms other competitors.
3. The paper is well written and easy to follow.

**Reasons To Reject:**

1. The proposed method outperforms other method, but more experiments are required. First, experiments on new entities should be conducted to evaluate the proposed method. One of the contributions is on inductive TKG reasoning. Detailed analysis on inductive reasoning is required.
2. To evaluate the proposed method, experiment on standard TKG setting would be helpful. The proposed method shows good results and how strong it is on standard TKG setting substantiates the superiority of the proposed method
3. It would be better to include other baselines such as RE-NET [Jin et al., 2019], RE-GCN [Li et al., 2021].

**Reproducibility:**

3: Could reproduce the results with some difficulty. The settings of parameters are underspecified or subjectively determined; the training/evaluation data are not widely available.

**Reviewer Confidence:**

4: Quite sure. I tried to check the important points carefully. It's unlikely, though conceivable, that I missed something that should affect my ratings.

---

> ### Author Rebuttal · Authors · 2023-08-29
>
> __Response to the Concerns of Reviewer dRYt.__
>
> We sincerely thank you for your valuable suggestions. We respond to your comments as follows
>
> > Question 1.  The proposed method outperforms other methods, but more experiments are required. First, experiments on new entities should be conducted to evaluate the proposed method. One of the contributions is on inductive TKG reasoning. Detailed analysis on inductive reasoning is required.
>
> __Response to Question 1__: Thanks for your comment.
>
> First, we clarify the distinctions and connections between our proposed lifelong TKG reasoning and the inductive TKG setting. Inductive setting treats new entities as simultaneous. Lifelong TKG reasoning is a multiple consecutive inductive TKG reasoning processes (as explained in lines 34-41 and illustrated in Figure 1) and thus naturally requires to consideration of new entities in experiments. Therefore, all of our experiments involve new entities. Figure 3 proves the validity of our model for lifelong TKG reasoning at the same time as it demonstrates our ability for inductive TKG setting.
>
> > Question 2.  To evaluate the proposed method, experiments on standard TKG setting would be helpful. The proposed method shows good results and how strong it is on standard TKG setting substantiates the superiority of the proposed method.
>
> __Response to Question 2__: Thanks for your suggestions. We clarify two reasons for not including experiments on the standard TKG setting.
>
> + Reason 1: Our proposed lifelong TKG reasoning is essentially different from the standard transductive setting.
>
> + Reason 2: It is also worth noting that there are many papers on inductive reasoning, such as RMPI[2], MorsE[3], SE-GNN[4], MBE[5], LKGE[6] and MaKEr[7], also do not conduct experiments in the standard transductive setting, since our model and all these works do not focus on the standard transductive setting.
>
>
> > Question 3. The authors provided the code for reproduction but I couldn't access the link.
>
> __Response to Question 3__: As the reviewer NDHZ said, "The code is available". Please copy "https://anonymous.4open.science/r/EMNLP2023_ID520" to a browser.

---

### Official Review · Reviewer_NDHZ · 2023-08-12

**Soundness:** 3

**Excitement:**

3: Ambivalent: It has merits (e.g., it reports state-of-the-art results, the idea is nice), but there are key weaknesses (e.g., it describes incremental work), and it can significantly benefit from another round of revision. However, I won't object to accepting it if my co-reviewers champion it.

**Paper Topic And Main Contributions:**

This paper proposes the temporal displacement, temporal-rule-based reward shaping and an edge-aware message passing module to solve the challenge of lifelong TKG extrapolation task. The authors extract three dedicated datasets suitable for lifelong TKG extrapolation tasks from existing datasets and the proposed method achieves state-of-the-art performance on these datasets.

**Questions For The Authors:**

A. In Line 136, ‘the harder entity prediction’, the expression is too vague and needs more explanation. \
B. In the abstract, ‘lifelong TKG reasoning’ is the task, not the issue. The issue may indicate the unseen entities and future timestamps.

**Reasons To Accept:**

A. This paper introduces a new setting for temporal knowledge graph reasoning, i.e., lifelong. \
B. The code is available.

**Reasons To Reject:**

A. The introduction to Temporal-Rule-Based Reward Shaping in Section 3.5 is not clear enough, and the authors are expected to expand on Eq. 9 further. \
B. According to the introduction, this method is actually to calculate the relative time of facts, and the idea of using the relative time to assist TKG reasoning has been proposed in TITer. What’s the difference between temporal displacement and relative time in TITer? \
C. The proposed temporal-rule-based reward shaping module seems an incremental method based on previous methods. \
D. About the first case study in Sec 5.5, it’s more like motivation and illustration, not a case study of your experimental result.

**Reproducibility:**

4: Could mostly reproduce the results, but there may be some variation because of sample variance or minor variations in their interpretation of the protocol or method.

**Reviewer Confidence:**

4: Quite sure. I tried to check the important points carefully. It's unlikely, though conceivable, that I missed something that should affect my ratings.

---

> ### Author Rebuttal · Authors · 2023-08-29
>
> __Response to the Concerns of Reviewer NDHZ.__
>
> We sincerely thank you for your valuable suggestions. We respond to your comments and provide clarifications as follows.
>
> > Question 1.  According to the introduction, this method is actually to calculate the relative time of facts, and the idea of using the relative time to assist TKG reasoning has been proposed in TITer. What’s the difference between temporal displacement and relative time in TITer?
>
> __Response to Question 1__: Thanks for your comment. We would like to clarify that the concept of calculating the relative time of facts in tasks related to TKGs has been widely adopted for a long time. Our contribution lies in the way our model leverages this concept (in various perspectives from TITer), rather than proposing it.
>
> > Question 2. The proposed Temporal-Rule-Based Reward Shaping module seems an incremental method based on previous methods.
>
> __Response to Question 2__: Thanks for your comment. We would like to make the following clarification.
>
> This module is one of several proposed modules in our model. First, we have accurately identified the research gaps in the technical weaknesses of existing RL-based KG reasoning works through a comprehensive examination. Secondly, we propose a new Temporal-Rule-Based Reward Shaping module that specifically addresses these research gaps. Thirdly, the experiments demonstrate the superiority of our proposed module, affirming the novelty of our contributions.
>
>
> > Question 3. In Line 136, ‘the harder entity prediction’, the expression is too vague and needs more explanation.
>
> __Response to Question 3__: Thanks for your comment. In line 136, we wrote "PathCon leverages relational message passing for relation prediction, however, we aim at the harder entity prediction.", which means that compared with the relation prediction task, the entity prediction task is more challenging. For better understanding, we will revise the writing to "PathCon leverages relational message passing for the relation prediction task, however, we aim at the entity prediction task, which is more challenging. Because there are more entities than relations."
>
> > Question 4. In the abstract, ‘lifelong TKG reasoning’ is the task, not the issue. The issue may indicate the unseen entities and future timestamps.
>
> __Response to Question 4__: Thanks for your suggestions.
>
> "Unseen entities and future timestamps" are the challenges in our proposed "lifelong TKG reasoning", which is a new issue. The task is TKGC.

---

### Meta-Review · Area_Chair_bUci · 2023-09-18

**Recommendation:** 3

**Metareview:**

The paper studied the lifelong temporal knowledge graph reasoning problem. The authors first constructed datasets for this task and proposed a model under the framework of RL.

All reviewers agreed that the lifelong TKG task is important and novel. Nevertheless, the reviewers also raised some concerns on the experiments setup and comparison. Most of these concerns were addressed by the authors during the rebuttal phase.

---

### Decision · Program_Chairs · 2023-10-07

**Decision:**

Accept-Findings

**Comment:**

The paper studied the lifelong temporal knowledge graph reasoning problem. The authors first constructed datasets for this task and proposed a model under the framework of RL.

All reviewers agreed that the lifelong TKG task is important and novel. Nevertheless, the reviewers also raised some concerns on the experiments setup and comparison. Most of these concerns were addressed by the authors during the rebuttal phase.